# Immune Cell Degranulation in Fungal Host Defence

**DOI:** 10.3390/jof7060484

**Published:** 2021-06-16

**Authors:** Adley CH. Mok, Christopher H. Mody, Shu Shun Li

**Affiliations:** 1Department of Microbiology Immunology and Infectious Diseases, Cumming School of Medicine, University Calgary, Calgary, AB T2N 4N1, Canada; adley.mok@ucalgary.ca; 2Calvin, Phoebe and Joan Snyder Institute for Chronic Diseases, Cumming School of Medicine, University of Calgary, Calgary, AB T2N 4N1, Canada

**Keywords:** granule, degranulation, trafficking, host defence

## Abstract

Humans have developed complex immune systems that defend against invading microbes, including fungal pathogens. Many highly specialized cells of the immune system share the ability to store antimicrobial compounds in membrane bound organelles that can be immediately deployed to eradicate or inhibit growth of invading pathogens. These membrane-bound organelles consist of secretory vesicles or granules, which move to the surface of the cell, where they fuse with the plasma membrane to release their contents in the process of degranulation. Lymphocytes, macrophages, neutrophils, mast cells, eosinophils, and basophils all degranulate in fungal host defence. While anti-microbial secretory vesicles are shared among different immune cell types, information about each cell type has emerged independently leading to an uncoordinated and confusing classification of granules and incomplete description of the mechanism by which they are deployed. While there are important differences, there are many similarities in granule morphology, granule content, stimulus for degranulation, granule trafficking, and release of granules against fungal pathogens. In this review, we describe the similarities and differences in an attempt to translate knowledge from one immune cell to another that may facilitate further studies in the context of fungal host defence.

## 1. Introduction

Fungal diseases are a tremendous medical problem. The frequency of fungal infections continues to climb, predominantly because of an increased number of immunocompromised and critically ill patients [1,2]. The mortality from invasive fungal infections is often greater than 50%, and it is estimated that more than 2.3 million people die from fungal infections each year [3]. Lack of effective therapy is largely responsible for the high mortality [4]. The therapeutic options for fungal infections are limited and associated with toxicities, which has led to an interest in immune therapeutic approaches [5]. One such therapeutic target is granule-dependent release of antifungal molecules used in host defence.

Immunity is a sophisticated, coordinated system consisting of highly specialized innate and adaptive immune cells that play vital roles against fungi. Both innate and adaptive immune cells are involved in fungal host defence, such as against organisms among the Ascomycota (*Aspergillus fumigatus*, *Candida albicans*), Basidiomycota (*Cryptococcus neoformans*), and Zygomycota (*Rhizopus oryzae*). NK cells, eosinophils, mast cells, neutrophils, and T cells boast intracellular membrane bound vesicles, which store compounds that can be immediately deployed for host defence. These intracellular compartments have been called “secretory vesicles”, “secretory lysosomes”, or “granules”. These organelles form when products of the trans-Golgi network are packaged into transport vesicles. Transport vesicles move the cargo to an endosome that undergoes acidification and processing of the cargo leading to formation of the secretory vesicles. Secretory vesicles contain molecules that induce fungal cell death or stasis when immune cells engage an invading pathogen.

Immune cells not only act independently, but also work in a complex manner by releasing factors and cytokines that signal and/or prime each other to effectively clear infections. Depending on the immune cells, granules are deployed in different ways. The immune cell can bind to the pathogen and antimicrobial compounds are released directly onto the pathogen. Alternately, immune cells can bind to another host cell that contains the pathogen. In this case, the antimicrobial compounds are released in a directed way through an immunological synapse (IS) between the immune cell and the host cell containing the pathogen, leading to death of the microbe. Immune cells may not bind directly to the pathogen, but receive signals from the pathogen or surrounding cells, causing release of antimicrobial compounds in a non-directional way in the vicinity of the pathogen. Finally, granules are released onto the pathogen surface when it is trapped in an extracellular matrix made up of DNA. Granules are also recruited to phagosomes that contains the engulfed pathogen, but this intracellular pathway will not be the subject of this review.

The mechanisms and machinery by which granules are trafficked within immune cells and released on to the pathogen vary depending on the immune cells and target pathogens. However, the immune cell subtypes share similarities in activation, signaling, and granule trafficking towards the plasma membrane. This review describes our current understanding of the granules in immune cells. We highlight the similarities in different characteristics and processes in granule development, content, storage, signaling, trafficking, release, and function in various immune cell types during host defence against fungal infections. Understanding these characteristics may allow for knowledge transfer between scientists working with different cells and may lead to insights into the development of immune-based therapeutics for different cells employing similar granule-mediated mechanisms.

## 2. Granule Characteristics in Different Immune Cell Subsets

Despite common features, secretory vesicles are described and classified differently for each immune cell. Granules are usually classified by size, morphology, and density using electron microscopy. If the buoyant densities of granules differ, they can be separated by centrifugation, which allows proteomic approaches to identify constituents. NK cells have three types of granules: type 1, type 2, and intermediate [6], which are grouped by their morphology (Table 1 and Figure 1, panel Aa, Ab, and Ac). Type 1 granules are 50–700 nm in diameter and filled with a dense core surrounded by a thin layer of vesicles [7]. Type 2 granules are 200–1000 nm in diameter and characterized by multiple vesicles and membrane whorls [6]. Intermediate granules have dense cores and multiple vesicles and are less abundant than type 2 granules [8]. Type 1 granules are fully mature while other types represent different stages of granule development [6]. Different components of the granules contain different constituents. The dense core contains cytolytic proteins, while the multivesicular domains contain lysosomal proteins (Table 2) [7]. By contrast, the granules of CD8+ T cells have not been separated by morphology. Rather, granules are characterized in one group with variable granule morphology that resembles the spectrum of granules in NK cells ranging from 100 to 1300 nm [9]. Granules in cytotoxic T cells can be separated by sucrose gradients (Table 1), which allows for separation of different proteins in granules of different buoyant density [10].

Granulocytes (neutrophils, eosinophils, and mast cells) have more than one type of granule and may contain different cytolytic contents (Figure 1, panel B, C, and D). Mast cell granules are distinguished by their membrane proteins and serotonin rather than their microscopic appearance (Table 1). Type I and II mast cell granules all contain proteins of the major histocompatibility complex (MHC) class II, β-hexosaminidase, lysosome-associated membrane protein (LAMP)-1 and 2, and mannose 6-phosphate receptor (M6PR), while type III granules lack MHC class II, LAMP-1, LAMP-2, and M6PR. Type I granules, in contrast to type II and III have serotonin [14].

Eosinophils have two types of granules: primary and secondary specific (Table 1) [15]. Sizes range from 500 to 1000 nm [16]. The secondary specific granules have a distinctive dense crystalline core that is surrounded by a less dense matrix and enclosed by a trilaminar membrane [17]. The primary granules are smaller than the secondary specific granules and lack a crystalline core [12]. Granules in all immune cell types appear as distinct, electron dense membrane bound intracytoplasmic organelles that can also be seen on light microscopy. Granules are of similar size (50–1300 nm), with most in the range of 200–500 nm.

Neutrophils have three types of granules: primary azurophilic, secondary specific, and tertiary gelatinase granules (Table 1). These granules are classified by their sizes and intensity on electron microscopy as well as their granule content. Primary granules are electron dense and range from 500 to 1000 nm [18]. Secondary granules range from 200 nm to 500 nm and tertiary granules have a mean diameter of 187 nm [18,19].

## 3. Cytolytic Contents of the Granules in Each Immune Cell Subtypes

The contents of granules in cytotoxic lymphocytes including NK cells and CD8+ T lymphocytes consists of cytolytic and pro-apoptotic proteins including perforin (cytolysin), granulysin, granzymes, the cationic protein LL 37, FasL, and lysosomal membrane glycoprotein LGP120 (Table 1) [7]. In T cells, different proteins have been isolated from different granule fractions. For example, perforin is isolated from one fraction separated by gradient centrifugation, while granzyme B and granulysin are in another (Table 1) [10].

Likewise, different granules in neutrophils and eosinophils contain different cytolytic molecules. Primary azurophilic granules in neutrophils contain proteases (neutrophil elastase, cathepsin G, proteinase 3), peroxidases (myeloperoxidase), and membrane cytolytic molecules (defensins and bactericidal/permeability-increasing protein (BPI) (Table 1) [20]. Secondary specific granules contain iron binding proteins (lactoferrins), membrane cytolytic molecules (defensins, BPI) and peroxidases (MPO). Tertiary gelatinase granules contain matrix metalloproteinases, sialidase, azurocidin, and lysozyme (Table 1) [20].

Primary granules of eosinophils contain galactin-10, which forms Charcot-Leyden crystals and secondary specific granules that contain eosinophil peroxidase (EPO), major basic protein (MBP), eosinophil cationic protein (ECP), and eosinophil-derived neurotoxin (EDN) (Table 1) [15,21].

While mast cells contain three types of granules, based on membrane proteins, they do not have differences in cytolytic content. These granules contain cytolytic molecules such as chymase, tryptase, mast cell carboxypeptidase A3 (CPA3), and granzymes [14]. Of note, CPA3 functions together with chymases and tryptases to degrade proteins and peptides [22]. There is evidence that secreted factors from mast cell granules mediate antifungal activity against *C. albicans*, although it is not known which factors are key to the process [23].

## 4. Mechanisms of Fungal Recognition, Activation, and Cytotoxicity

Granule trafficking is activated when immune cells recognize the fungal pathogen through pathogen associated molecular patterns through their pattern-recognition receptors (PRRs), NK cell activating receptors, or the T-cell receptor (TCR). Ligands for PRRs are usually the components (e.g., carbohydrates) of the fungal cell wall, or fungal-derived RNA or DNA, which are not found in mammals. The PRRs include C-type lectin receptors (CLRs), toll-like receptors (TLRs), nucleotide binding and oligomerization domain (NOD)-like receptors (NLRs), and retinoic acid inducible gene (RIG)-like receptors (RLRs). NK cells have activating receptors that function as PPRs, such as NKp30 [24], NKp46 (natural cytotoxicity receptor-1), and CD56 for fungal recognitions [25,26]. These receptors are required for activation by *Cryptococcus, Candida, and Aspergillus* [27].

Fungal carbohydrates, such as β-glucan, α -glucan, and chitin are well-known ligands of the fungal cell wall that are recognized by innate immune cells, such as NK cells, neutrophils, mast cells, eosinophils, and macrophages [27]. CD8+ T cells recognize fungi via the TCR by cross-presentation where fungal antigens are presented on the major histocompatibility complex-1 (MHC-1) on antigen presenting cells, such as dendritic cells after phagocytosing the fungal target cell [28]. A signaling cascade is activated, and granules are trafficked to the target cells upon their recognition.

## 5. Signal Transduction Leading to Degranulation

The signaling pathway leading to degranulation involves a complex network of pathways that orchestrate many responses that ultimately lead to degranulation. The main signaling mechanisms include immunoreceptor tyrosine-based activation motifs (ITAMs)/Syk ➔ integrins, G-proteins and MyD88 (Table 2 and Figure 2). The signaling pathway of NK cells operate downstream of ITAM-dependent and ITAM-independent motifs activated by redundant Src family kinases [29,30]. Syk-1 then initiates activation of two signal pathways: the phosphoinositide-3 kinase (PI3K) ➔ extracellular receptor kinase (ERK)2 and the phospholipase Cγ (PLCγ) ➔ JNK1 pathways [31,32], through which polarization of the microtubule organizing center (MTOC) and cytolytic granules occur before subsequent degranulation of the cytotoxic granules into the target cell [6,33,34,35]. Additionally, *Cryptococcus* concurrently activates the β1-integrin pathway leading to activation of integrin linked kinase (ILK) ➔ Ras-related C3 botulinum toxin substrate (Rac) ➔ PI3K ➔ Erk [36]. Here, activation of both Src family kinases and Rac are required during fungal mediated degranulation [36]. For CD8+ T cells, this process is mediated by engagement of the TCR/CD3 complex, which lead to a signaling cascade utilizing Src family kinases, such as LymphoCyte-specific protein tyrosine Kinase (LCK), which phosphorylate ITAMs ➔ Zeta-chain-Associated Protein kinase 70(ZAP70) ➔ Linker for Activation of T cells (LAT)/PLCγ/interleukin-2-inducible T-cell kinase (ITK) ➔ Phosphatidylinositol 4,5-bisphosphate (PIP2) ➔ Inositol triphosphate (IP3) ➔ Ca^2+^ influx ➔ degranulation [37]. Another series of signaling cascades is required during granule trafficking for MTOC polarization. MTOC polarization signaling starts after PIP2 is hydrolyzed leading to the formation of lipid second messenger’s diacylglycerol (DAG). This in turn leads to synaptic recruitment of protein kinase C (PKC) isoforms, which leads to MTOC and granule polarization to the plasma membrane [37]. Mast cell degranulation is achieved when their PPRs (TLR4), cytokine receptors (CCR1), c-KIT, or FcεRI are engaged. The corresponding ligands for these PRRs respectively are HSP70, CCL3, stem cell factor (SCF), and IgE. The receptors signal via a variety of pathways that lead to phosphorylation of LAT [38]. Phosphorylated-LAT triggers PLCγ and PI3K phosphorylation followed by a chain of downstream signaling protein phosphorylation, resulting in degranulation [38]. Interestingly, the central role of PLCγ bears similarities to that of NK and CD8+ T cells in mediating degranulation and trafficking, although there are differences.

Granule trafficking and degranulation in eosinophils is initiated following the engagement of chemokine receptor 3 (CCR3), leading to the association of G-protein leading to the MAPK pathway and then the subsequent activation of PI3K. This precedes AKT and ERK activation, which modulates the activation of downstream effectors that leads to microtubule reorganization and subsequent granule degranulation [39]. Additionally, β2-integrin has been implicated in eosinophil degranulation [40].

In neutrophils, the signaling mechanism starts with the activation of neutrophils through surface receptors (PRRs, GPCRs, FcRs, selectins, and integrins) that triggers the activation of a kinase cascade (Src family kinases and protein signaling kinase 2 (Pyk2) [13]. Central downstream effectors of these kinases target cytoskeletal remodeling. These central effectors include Vav, PLC, and PI3K, which activates Rac and paxillin. This facilitates microtubule polarization, and the generation of PIP3, which facilitates polarization and actin remodeling at the degranulation site [13].

## 6. Granule Trafficking Leading to Degranulation

Granule mobilization and trafficking varies in different immune cells, however, there are general schemes that are common among all cells (Table 3). Initially, granules begin to be recruited along the microtubule when activated. Kinesins and dyneins are key microtubule transport proteins that function as motor proteins in the positive and negative direction, respectively, to traffic the granules either by moving the microtubules with an attached granule or by moving the granule on the microtubule. Some kinesins organize the microtubule network, while others are responsible for cargo transport. Examples of these kinesins, respectively, are Eg5-kinesin and kinesin-1. Eg5 kinesin has an extensive role in microtubule stabilization and cross-linking, where it mediates microtubule movement [41]. By contrast, kinesin-1 (Table 3) is a cargo binding motor protein that is responsible for terminal transport of the granules to the cell membrane and has been characterized in CD8+ T cells [42], NK cells [43], and mast cells [44]. The initial phase of granule mobilization is mediated by dyneins where it directs the granules to the minus-end of the microtubules and towards the MTOC. The MTOC is then polarized by the action of dynein, with which the granules are polarized mediated by kinesin-1 towards the synapse before degranulation. The degranulation step of the granules at the terminal where they dock and fuse with the plasma membrane is regulated by a family of soluble N-ethylmaleimide sensitive factor (NSF) attachment protein REceptors (SNAREs) [45]. Here, the MTOC docks at the terminal where granules can move to the synapse and degranulate. The degranulation process is mediated by SNAREs, which are primary mediators of membrane docking and fusion. The vSNARE on the vesicle binds to the tSNARE on the target membrane allowing Rab to hydrolyze GTP, locking the tSNARE and vSNARE together, allowing for fusion of the vesicle and plasma membrane and release of granule contents.

## 7. Function of Granule Proteins

### 7.1. Perforin

Perforin is a pore-forming cytolytic protein that is expressed by NK, CD8+ T cells and in one report by neutrophils (Table 1) [46,47]. When released, perforin inserts into the target cell membrane and oligomerizes in a Ca^2+^ dependent manner to form pores or damage the cell membranes (Table 4) [48]. The resultant pores allow for entrance of other granule proteins to enter the target cells. As such, perforin plays a key role in the cytolytic process in targets, such as tumors and fungi. In fungal studies, evidence showed that perforin is required in NK cell antifungal activity against *C. neoformans* [34]. While in CD8+ T cells, granulysin rather than perforin is required during antifungal activity against *C. neoformans* [49].

### 7.2. Granzymes

In humans, immune cells can express five types of granzymes: granzymes A, B, H, K, and M [50]. These are serine proteases that cleave proteins inducing cell death [50,51,52]. In mammalian cells, granzymes activate a cascade of enzymatic cleavages leading to apoptosis or cleavage of proteins that are critical for viability (Table 4) [50]. In bacteria, granzymes disrupt electron transport leading to superoxide anion and thwarting bacterial oxidative defences [53]. Granzymes often work in conjunction with perforin. Perforin causes pores or damages the target cell membrane, which enables granzymes to enter and exert their cytotoxic effect on tumor cells. While the functions of granzymes are well known in tumor target killing, little is known about their role in killing of fungal cells. Other immune cell subtypes also contain granzymes including neutrophils [54], eosinophils [55], mast cells [56], which play a role in inflammation, antitumor activity, and antibacterial activity. Interestingly, since most granule-containing immune cell subtypes express granzymes, it is a common cytolytic molecule utilized for a wide variety of responses including fungi.

### 7.3. Other Proteins Causing Membrane Permeability

Defensins have long been characterized as antimicrobial compounds against fungal pathogens such as *Candida* albicans [57]. These cationic molecules are predominantly expressed by neutrophils but can be found in NK cells and CD8+ T cells [58]. The function of defensins is to permeabilize the pathogen membrane and inhibit cell wall synthesis that leads to cytolysis of target cell [59,60]. Their ability to permeabilize membranes is similar to that of granulysin and perforin. The pores formed by defensins are approximately 25 nm [60], which is large enough to permit cytolytic molecules such as granzymes which are 2.5 nm in stokes radius to cross the plasma membrane [61].

LL-37 is an antimicrobial molecule of the cathelicidin family that is found in neutrophils, NK cells, and CD8+ T cells (Table 1 and Table 4). LL-37 acts on the lipoprotein membranes of pathogens leading to membrane damage, and therefore works similarly to perforin, granulysin, and defensins by affecting the cell membrane and the cell wall of *C. albicans* [62].

Bacterial Permeability Inducing protein (BPI) is a 50 kDa protein that binds to lipopolysaccharides found on the cell membrane of gram-negative bacteria leading to permeability of the membrane and cell death [63]. Its antibacterial action is due to its displacing outer membrane calcium and magnesium ions, leading to membrane permeability [63]. BPI also displaces divalent cations, which perturbs arrangement of LPS molecules and results in membrane rupture [63]. BPI is present in neutrophils and eosinophils [63], but its function in fungal killing is unknown.

### 7.4. Proteases—Neutrophil Elastase (NE), Cathepsins, Proteinase 3, Matrix Metalloproteinases

NE is a key effector molecule in neutrophils during fungal infections (Table 1). NE deficiency can result in impaired killing of *Candida* and *Aspergillus* [64]. This molecule is closely related to the family of serine proteases, and cleaves proteins in the extracellular matrix and, thereby, facilitate fungal killing by regulating extracellular trap formation (Table 4) [65]. Further details of this mechanism are discussed in later sections. Cathepsin G is a serine protease that is known to be able to degrade extracellular matrix components and may have antimicrobial specific function (Table 4) [66]. The role of NE and cathepsin G is made evident by mice lacking NE or cathepsin G, which are more susceptible to fungal infections [64]. Proteinase 3 is an enzyme that processes LL-37 to its active form after neutrophil activation [67]. Proteinase 3 also has antimicrobial properties that are independent of its protease activity against *C. albicans*, but the mechanism of action is unknown [65]. Matrix metalloproteinases (MMPs) are involved in cleavage or extracellular matrix components such as gelatin and collagen [68]. They also play a role in leukocyte migration [68].

Secretory leukocyte protease inhibitor (SLPI) is an antiprotease that is highly enriched in the secondary granules of neutrophils (Table 4). It has been reported that SLPI has antifungal activity to *Aspergillus fumigatus* and *Candida albicans* [69].

### 7.5. Oxidative Agents: Myeloperoxidase (MPO) and Eosinophil Peroxidase (EPO)

MPO is a key effector molecule in neutrophils during fungal infections. MPO acts by converting H2O2 and halide ions to reactive oxygen species, which are highly toxic to microbes (Table 4) [70]. EPO is the major oxidative enzyme of eosinophils and functions like MPO by catalyzing the formation of oxidizing agents, which converts H_2_O_2_ + halide (Cl^-^) into hypochlorous acid (HOCl), which is toxic to microbes, such as the bacteria *Mycobacterium tuberculosis* (Table 4) [71]. The evidence that EPO has a cytolytic role against fungi is limited [72].

### 7.6. Iron Scavengers

Lactoferrin is a constituent of granules that is an iron-binding protein that is known for its iron-sequestering function that prevents iron uptake in *Candida* and *Cryptococcus*, leading to cell death [73,74], and inhibition of *A. fumigatus* conidia germination [75].

### 7.7. Alarmins: Azurocidin and Eosinophil-Derived Neurotoxin (EDN)

Azurocidin is a protein in azurophilic granules that is known for its antimicrobial functions by binding to the cell surfaces of the target cell as well as acting as a chemotactic agent for monocytes and macrophages (Table 4) [76]. EDN is released in response to *Alternaria* and *Penicillium* [77] and is known to be cytotoxic against helminth parasites and have antiviral activity due to its ribonuclease activity but its functions against fungi is unknown (Table 4) [78,79].

### 7.8. Iron scavengers

Lysozyme exerts it antimicrobial activity through damaging the cell wall of fungi by hydrolyzing the β1–4 glycosidic bond between N-acetylmuramic acid and N-acetylglucosamine, which are structural components of fungal chitin (Table 4) [80].

### 7.9. Major Basic Protein (MBP) and Eosinophil-Cationic Protein (ECP)

MBP and ECP are effector molecules known to have antifungal roles against *Alternaria alternata* [77]. MBP functions by disrupting the lipid bilayer membrane, resulting in cell damage [81]. ECP is a ribonuclease that binds to the cell wall and cell membrane. Although the mechanism is unknown in fungi bacterial ECP binds and destabilizes membranes which would result in cell death (Table 4) [82].

## 8. The Cells That Degranulate

### 8.1. NK Cells

NK cells are cytotoxic lymphocytes that degranulate to induce cell death. In addition to being cytotoxic for tumor cells, NK cells can also kill fungi. NK cell cytotoxicity is activated through ligation of the activating receptor with the respective ligand expressed by *Cryptococcus*, *Candida,* and *Aspergillus* leading to phosphorylation of signaling molecules downstream of the receptor [83]. NK cells mobilize perforin-containing granules along the microtubules in a stepwise manner in response to *Cryptococcus* prior to degranulation (Figure 3) [35]. Initial trafficking is characterized by granule movement away from the MTOC and immune synapse (counter-convergence and counter-polarization, respectively). This is followed by movement of the granules toward one another (congregation), movement toward the MTOC (convergence) and polarization of the MTOC and granules to the synapse (Figure 3) [35]. The events leading to polarization and position of granules at the synapse are unique to NK cell killing of fungi. These various trafficking events are orchestrated by the microtubules and the MTOC as well as kinesins and dynein [35,84]. Eg5 kinesin is a motor protein involved in mitosis. In NK cell response to *Cryptococcus*, Eg5 kinesin is responsible for initial granule counter-convergence and MTOC counter-polarization [35], which along with dynein are required for granule congregation, convergence, and polarization [35]. Once polarized to the IS, granules are released into the IS where perforin and other cytolytic molecules such as granzymes enter the target fungal cell. Perforin is a key effector molecule that NK cells use against *Cryptococcus* and *Candida* [24,85]. Perforin is presumed to make pores on the membrane through which granzymes and other cytolytic molecules enter into the target as it does in tumor cells [46]. Whether granzymes or other cytolytic molecules contained within the granules play a role in fungal killing is unknown. Kinesin-1, a cargo transport motor protein, is involved in the terminal granule transport to the IS in tumor targets [43], but its role in anti-fungal granule trafficking remains unknown. At the IS, granule exocytosis is mediated by syntaxin-11 which is an atypical Q-SNARE [45]. Syntaxin-11 is transported to the IS before granules fuse where it helps granules dock, prime, and fuse at the cell membrane. This is crucial for NK cell degranulation and without which NK cells failed to kill tumor cells [86]. While the function of syntaxin-11 is essential during NK cell antitumor cytotoxicity, its role in NK cell antifungal activity has not yet been demonstrated.

### 8.2. CD8+ T Cells

CD8+ T-cells play a major role in controlling fungal infections [87], however, direct killing is still not well understood. Currently, it has been shown that CD8+ T cells use granulysin, a pore-forming antimicrobial molecule, to kill *C. neoformans* [49]. In addition, yeast specific CD8+ T cells show a non-classical expression of granulysin and granzyme K instead of the classical perforin and granzyme B cytotoxic granule profile [88]. The mechanics of the granule-mediated cytotoxic activity by CD8+ T cells against fungal targets has not been explored. However, there may be similarities to the CD8+ T cell granule-mediated anti-tumor mechanism. The granule-mediated cytotoxicity is activated through two main receptors, T-cell receptor (TCR), and toll-like receptor (TLR)/scavenger receptor [89]. While TCR is the main receptor involved in the whole activation and degranulation signaling cascade, TLRs augment the activation through dendritic cell cross-presenting fungal antigens. While TCR is important for CD8+ T cells in antigen recognition of fungal pathogens, TLRs are used by CD8+ T cells in augmenting its activation during antigen cross-presentation [89]. Sensing of RNA via TLR3 promotes cross presentation by dendritic cells to class I-restricted CD8+ T cells. This is made evident by TLR3-/- mice, which are more susceptible to *Aspergillus* infection compared to control mice [89,90]. The significance of TLRs in CD8+ T cell recognition and activation is further exemplified when patients with deleterious mutations in TLR3 and TLR4 were more susceptible to various fungal infections [89,91,92,93]. Like NK cells, CD8+ T cells deliver their cytotoxic granule contents into the tumor IS after trafficking to the synapse with the help of microtubules and various motor proteins. Here, kinesins and dyneins are responsible for plus- (away from the MTOC) and minus- (toward the MTOC) end directed movement along microtubule tracks and aids in the MTOC polarization [94]. Dynein is responsible for minus end directed movement that pulls the MTOC toward the IS for successful degranulation at the IS [94]. Kinesins play an important role in plus-end directed movement of granules on the microtubules [95]. Specifically, terminal transport of lytic granules to the tumor IS is mediated by the kinesin-1/Slp3/Rab27a complex where kinesin-1 is essential for the final terminal movement of granules toward the IS [42]. Kinesin-4 was found to be important in regulating microtubule growth to allow for rapid remodeling and polarization of the microtubules when CD8+ T cells are in contact with antigen presenting cells [96]. However, the role of kinesin-4 is unclear in CD8+ T cell cytotoxic mechanisms. As for granule exocytosis at the IS, syntaxin-11 plays the same role as in NK cells during cytotoxic events.

### 8.3. Mast Cells

Mast cells are granulocytes that are well known for their role in allergic responses, but they also play a role in host defence against pathogens, including fungi. These cells localize at mucosal sites and are found in close contact with epithelial cells and venules and, as such, are considered tissue-resident sentinel cells [97]. Activation of mast cells during allergic responses occurs through IgE receptor-ligand interaction; however, against fungi, activation is achieved through other receptors such as TLR4 and CCR1. In contrast to cytotoxic lymphocyte degranulation, mast cell degranulation occurs at multiple degranulation sites along the plasma membrane and can engage multiple targets at once [98,99]. Mast cells also have directed degranulation via an IS when in contact with IgE and IgG targeted microbial pathogens [99]. A similar observation was made with *C. albicans* [97]. Although the study did not demonstrate direct degranulation, it did show that mast cells rearranged the α-tubulin cytoskeleton at the synapse and recruited LAMP1+ vesicles at the synapse with the fungus. The authors speculated that degranulation occurs at the IS where the contents are released on to the surface of the fungal cell that is exposed to the IS [97]. Further, the authors observed that mast cells tightly interact with the fungal cell such that it looks like it is engulfing the fungal cell [97]. This interaction is a broad-based invagination reminiscent of a phagocytic cup, in contrast to cytotoxic lymphocytes [97]. The tight phagocytotic-like interaction occurred with the hyphal, but not the yeast form of *C. albicans* [97], and this interaction was not a result of pathogen invasion of the mast cells but rather actin dynamics of mast cells [97].

The differences between the IS of cytotoxic lymphocytes (NK and CD8+ T cells) and mast cells may provide important insights in mechanisms of degranulation. While mast cell granule trafficking requires microtubules, the polarization of granules does not require MTOC or Golgi apparatus. Instead, polarization requires cytoskeletal rearrangement where extension of microtubules is sufficient to guide granules to the plasma membrane [99,100]. However, these observations were made in studies with *Toxoplasma gondii* and it remains unknown if this is recapitulated with fungal pathogens [99]. Dyneins are motor proteins involved in granule trafficking, while kinesin-1 moves granules in the positive direction to the membrane and facilitates degranulation. Dynein, complexed with Rab-interacting lysosomal protein (RILP) mediate the retrograde transport of granules along the microtubule network [101]. Like CD8+ T cells and NK cells, degranulation is controlled by kinesin-1 in mast cells [44]. It is not known, however, if kinesin-1 participates in the mast cell response to fungi. Granule exocytosis is enabled by a complex of proteins consisting of synaptosome-associated protein (SNAP)-23, syntaxin-4 (a target(t)-SNARE), and vesicle-associated membrane protein 8 (VAMP-8) [102]. Similar to syntaxin-11 in CD8+ T cells and NK cells, these proteins form complexes and modulate granule exocytosis at the IS [102].

### 8.4. Eosinophils

Like mast cells, eosinophils are well known in allergic responses, but also have a role against fungal infections. Indeed, eosinophilia is often seen in fungal infections [103]. Eosinophils are known to interact with fungal pathogens, such as *A. alternata*, A. fumigatus, and *C. albicans* [77]. The role of kinesins and dyneins in eosinophil microtubule reorganization and granule trafficking is unknown but kinesins and dyneins play a common role in microtubule and cytoskeleton reorganization. Whether they are key to successful granule trafficking and degranulation in eosinophil antifungal activity is unknown. Syntaxin-17 is a SNARE that was found on the granules of eosinophils and is suggested to be involved in membrane trafficking although further characterization of syntaxin-17 is needed [104]. Whether other membrane fusion machinery is involved remains known.

Unlike NK cells and CD8+ T cells that mainly kill targets by direct degranulation, eosinophils kill target cells by utilizing the granule in various ways: classical exocytosis (as in mast cells) or compound exocytosis, cytolysis, or piecemeal degranulation [103]. Compound exocytosis constitutes the fusion of specific granules with each other and with the plasma membrane for granule content release. Cytolysis constitutes the release of intact granules into extracellular milieu. Piecemeal degranulation is a secretory process in which small packets of granules are selectively mobilized to the plasma membrane, fuse, and the granule contents are sequentially released [105]. Piecemeal degranulation is also seen in mast cells and cytotoxic lymphocytes. Eosinophils have been shown to respond to *A. alternata* by releasing granule contents via compound exocytosis in which vesicles fuse with one another, and by releasing extracellular DNA traps along with release of granules in response to *A. fumigatus*. There is much to be done to understand the role of degranulation in the role of eosinophils in response to fungi.

### 8.5. Neutrophils

Neutrophils play a critical role in host defence against fungi, such as Aspergillus spp., Candida spp., and Blastomyces spp. [106,107,108]. They are one of the largest immune cell populations and one of the first responders to pathogenic invasion. These cells employ three different mechanisms to kill: (1) phagocytosis, (2) degranulation, and (3) NETosis. In the case of direct killing by degranulation, granules are transported through actin and microtubule reorganization from cytosol to the plasma membrane [109,110]. The degranulation process at the cell membrane is mediated by two SNARE complexes, made up of syntaxin 4/SNAP-23/VAMP-1 and syntaxin 4/SNAP-23/VAMP-2, during exocytosis of specific and tertiary granules [111]. Degranulation of primary azurophilic granules is mediated by a complex of syntaxin 4 and VAMP-1/VAMP-7 [111]. Granules are also transported in the same way to the phagosomes by actin cytoskeleton remodeling and microtubule assembly during phagocytosis [112]. While movement of granules along microtubule tracks is believed to be mediated by dynein and kinesin motor proteins, there is limited evidence [113,114]. Further, it remains unclear of how granules are trafficked in neutrophils against fungal pathogens. NETosis is a process where neutrophils undergo lysis where a matrix made of DNA or neutrophil extracellular traps (NET), traps the pathogen followed by release granule contents containing “azurosomes”, consisting of eight azurophilic cytolytic granule proteins [110]. The azurosomes form on the granule’s inner membrane in response to NET-inducing stimuli [115]. Secretion of the cytolytic molecules into the extracellular milieu where the fungal pathogen is trapped within the NET would result in death [115].

## 9. Conclusions

While immune cells deploy secretory vesicles for direct cytotoxic activity against fungal pathogens, they also employ the contents of secretory vesicles to signal a vast array of other responses that lead to the ultimate clearance of the fungal invasion. In this review, we discussed the diversity in granule-mediated mechanisms by the NK cells, CD8+ T cells, mast cells, eosinophils, and neutrophils. Further studies are needed to fully understand granule contents and detailed mechanisms used in response to fungal invasion in different immune cells.

## Figures and Tables

**Figure 1 jof-07-00484-f001:**
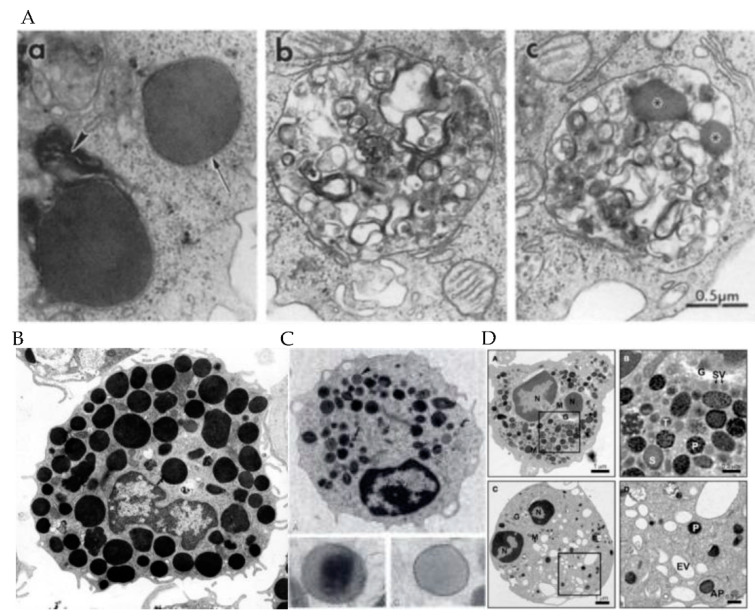
**Panel A**: Electron micrographs showing the heterogeneity of RNK-16 (NK cell) granules. (**A**a) Type I granules. (**A**b) A type II granule. (**A**c) An intermediate granule with small cores (figure from [7] with permission). **Panel B**: rat mast cells with dark electron-dense granules (figure from [11] with permission). **Panel C**: cytoplasmic granules of eosinophils include many membrane-bound, large, dense, spherical, crystalloid-containing granules; less numerous, large, dense, spherical, crystalloid-free granules (figure from [12] with permission). **Panel****D**: neutrophil morphology visualized by electron microscopy. (Top left, top right) The cytosol of a resting cell is filled with vesicles, with primary granules (P) staining intensely dark with diaminobenzidine, while secondary (S) and tertiary (T) granules show more translucent staining. Secretory vesicles (SV) are near the Golgi complex (G). Few mitochondria (M) are observed. (Bottom left, bottom right) (figure from [13] under Creative Commons Attribution License (CC BY)).

**Figure 2 jof-07-00484-f002:**
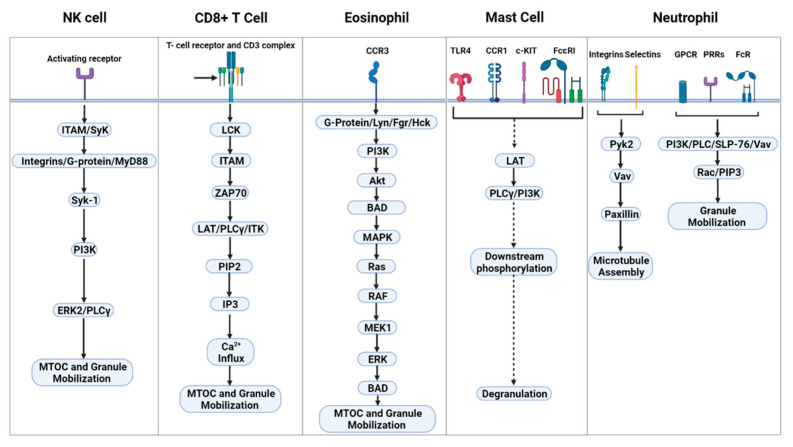
Canonical signaling pathways of granule trafficking.

**Figure 3 jof-07-00484-f003:**
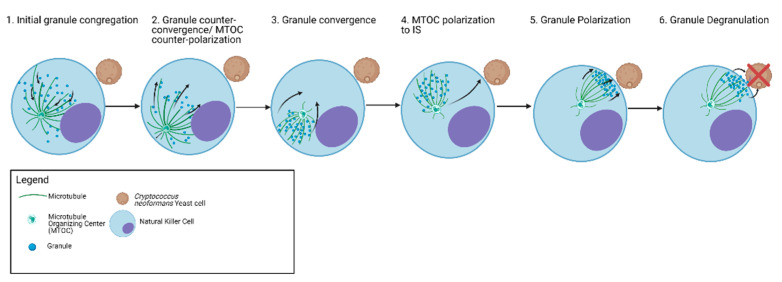
Graphical representation of Stages of granule trafficking during an NK cell mediated cytotoxic event against fungal pathogen *C. neoformans*.

**Table 1 jof-07-00484-t001:** Granule types and contents in various immune cells.

	NK Cells	CD8+ T Cells	Mast Cells	Eosinophil	Neutrophil
**Types**	**Type 1 Granule (fully Formed)** 50–700 nm Contains a dense core surrounded by thin layer of vesicles **Type 2 Granule** 200–1000 nm Contains multiple vesicles and membrane whorls **Intermediate Granule** Contains dense cores and multiple vesicles, less abundant than type 2 granules	**Cytotoxic****Granule** 100–1300 nm Exists in tiny droplets, dark-core bodies surrounded by a thin membrane, or large granules containing small internal vesicles	**Type 1 Granule** MHC class II, β-hexosaminidase, lysosomal membrane protein (LAMP)-1/2, Mannose- 6-phosphatereceptors (M6PR) **Type 2 Granule** MHC class II, β- hexosaminidase, LAMP-1/2, M6PR, Serotonin **Type 3 Granule** β-hexosaminidase, serotonin	**Primary Granule****:** 500–1000 nm Lack crystalline core **Secondary (Specific) Granule****:** 500–1000 nm Contain distinctive dense crystalline core that is surrounded by a less dense matrix and enclosed by a trilaminar membrane	**Primary Azurophilic Granule** electron dense 500–1000 nm **Secondary Specific****Granule** 200–500 nm **Tertiary (gelatinase)****Granule** Mean size of 187 nm
**Content**	In all granule types: Perforin Granzymes Defensins 1–3 LL-37 Granulysin FasL and TRAIL	In all granule types: Perforin Granzymes Defensins 1–3 LL-37 Granulysin FasL and TRAIL May be separated by granule density	No distinct difference in content between granule types but are: chymase, tryptase, mast cell carboxypeptidase A3 (CPA3), β-hexosaminidase, histamine, granzyme	**Primary Granule****:** Charcot–Leyden crystal protein (galactin-10) **Secondary Granule****:** eosinophil peroxidase (EPO) major basic protein (MBP) eosinophil cationic protein (ECP) eosinophil-derived neurotoxin (EDN)	**Primary Granule****:** neutrophil elastase, myeloperoxidase (MPO), defensins, cathepsin G, proteinase 3 **Secondary Granule****:** lactoferrins, defensins, BPI, MPO, lysozyme, LL-37 **Tertiary Granule****:** matrix metalloproteinases, azurocidin, lysozyme

Contents listed are an up-to-date comprehensive list of molecules necessary for cell death.

**Table 2 jof-07-00484-t002:** Pathways and modes of degranulation in various immune cells.

	NK Cells	CD8+ T Cells	Mast Cells	Eosinophil	Neutrophil
**Pathway**	ERK2 ➔ JNK1 ➔ MTOC, granule polarization and cytotoxicity ITAM dependent and independent signaling ➔ MAPK cascade ➔ NK cell effector functions	TCR ➔ LCK/ZAP70 ➔ LAT/PLCγ/ITK ➔ PIP2 ➔ IP3 ➔ Ca2+ influx ➔ degranulation	Surface receptors (CCR1, TLR4, KIT, or FcεRI). G-protein, MyD88, Jak/STAT, ➔ Lck-phos ➔ LAT-phos ➔ PLCγ ➔ degranulation	CCR3 ➔ G-protein/Lyn, Fgr, Hck ➔ PI3K ➔ Akt ➔ BAD ➔ MAPK ➔ Ras ➔ RAF ➔ MEK1 ➔ ERK ➔ BAD	Microtubule assembly: selectins/integrins ➔ Pyk2 ➔ Vav ➔ paxillin granule mobilization: surface receptors (GPCR, Fc-R, PPRs) ➔ PI3K/PLC/SLP-76/Vav complex ➔ Rac and PIP3
**Mode**	Cytotoxic degranulation through direct contact of target cells	Cytotoxic degranulation through direct contact of target cells	Anaphylactic/cytotoxic degranulation Phagosomal granule fusion and degranulation	Piecemeal degranulation Intact granule exocytosis and EETosis Phagosomal granule fusion and degranulation	Cytotoxic degranulation Phagosomal granule fusion NET formation and degranulation onto NETs

**Table 3 jof-07-00484-t003:** The roles of microtubules, dyneins, kinesins, and SNAPs/SNAREs in granule trafficking in various immune cells.

	NK Cells	CD8+ T Cells	Mast Cells	Eosinophil	Neutrophil
**Microtubules**	Microtubules facilitate the delivery of lytic granules to the synaptic cleft between NK cells and target cells through microtubule associated motor proteins	Microtubules facilitate secretory granule dynamics and degranulation by microtubule protrusion formation and reorganization	Microtubules serve as scaffold for Kinesin and Dynein	Microtubule Reorganization facilitates granule release	Granules are recruited and mobilized by microtubules
**Dynein/Kinesin**	Dynein mediates minus directed movement of granules to converge to MTOC Kinesin-1 mediates terminal granule movement and degranulation to IS Eg5 Kinesin is involved in NK cell granule trafficking during antifungal activity	Terminal transport of lytic granules is mediated by the kinesin-1/Slp3/Rab27a complex Kinesin-4 KIF21B limits microtubule growth to allow rapid centrosome polarization in T cells. Lytic granules have kinesin-dependent motility on microtubules in vitro	Kinesin-1 controls mast cell degranulation through PI3K-dependent recruitment to the granular Slp3/Rab27b complex Dynein is involved in retrograde transport of secretory vesicles	Role of motor proteins are unknown	Kinesins are involved in granule-microtubule interactions and movement
**SNAPs/SNAREs**	atypical Q-SNARE syntaxin 11/Sec/Munc (SM) family Mediate granule exocytosis by providing the tight complex that brings the granule to the cell membrane and enabling for granule fusion	atypical Q-SNARE syntaxin 11/Sec/Munc (SM) family Mediate granule exocytosis by providing the tight complex that brings the granule to the cell membrane and enabling for granule fusion	SNARE proteins function to mediate constitutive trafficking events through both endocytic and secretory pathways	Specific membrane docking of granules through interaction with plasma membrane t-SNARES, SNAP-23, and syntaxin-4 Qa SNARE (Syntaxin17) is involved in granule transport	Two SNARE complexes, made up of syntaxin 4/SNAP-23/VAMP-1 and syntaxin 4/SNAP-23/VAMP-2, are involved in the exocytosis of specific and tertiary granules Interactions between syntaxin 4 and VAMP-1/VAMP-7 are involved in the exocytosis of azurophilic granules.

**Table 4 jof-07-00484-t004:** Classification of proteins found in granules.

Classification of Granule Proteins	Proteins
Cytolytic, Cell wall and Membrane disrupting/pore forming	Perforin, granulysin, defensins, LL-37, eosinophil cationic protein, major basic protein, bactericidal/permeability-increasing protein, azurocidin
Peptidoglycanases	lysozyme
Protease Inhibitors	Secretory leukocyte protease inhibitor (SLPI)
Immune modifying/cytokines	IFN-γ, TNF-α, GM-CSF, VEGF IL-1a, IL-10, IL-2, IL-3, IL-4, IL-5, IL-6, IL-10, IL-12, IL-13, IL-16, IL-17A, IL-17F, IL-21, IL-22, IL-25, IL-27
Oxidative agents	MPO, EPO
Pro-apoptotic agents (serine proteases including tryptases and chymases)	Mast cell tryptase and chymases, CPA3, granzymes A, B, H, K, M,
Chymotrypsin-like serine proteases	Neutrophil elastase, cathepsin G
Iron binding proteins	Lactoferrin
Extracellular matrix degrading Matrix Metalloproteinases/Gelatinases	MMP-8, MMP-9
Ribonucleases including Cationic proteins	ECP

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
