# Peer review of "Immune Cell Degranulation in Fungal Host Defence"

_jof, 2021, doi:10.3390/jof7060484_

Round 1

Reviewer 1 Report

I applaud the authors' effort to synthesize the current knowledge on immune cell degranulation in a more systematic and logically organized manner. I do have some questions which I have noted in the attached annotated PDF. My primary discomforts are about the readabilities (or difficulties therein) of the Table 1 and the descriptions of the pathways. The value of a review such as this lies in having the reader grasp the minute details of the processes described; currently, it's too dense. Perhaps the authors could expend a bit more effort towards ensuring a better readability?

Author Response

General comment 1: I applaud the authors' effort to synthesize the current knowledge on immune cell degranulation in a more systematic and logically organized manner.

Response: We thank the reviewer for the encouraging comment.

General comment 2: My primary discomforts are about the readabilities (or difficulties therein) of the Table 1 and the descriptions of the pathways. The value of a review such as this lies in having the reader grasp the minute details of the processes described; currently, it's too dense. Perhaps the authors could expend a bit more effort towards ensuring a better readability?

Response: We thank the reviewer for this comments, which allowed us to make this review easier to follow. We have rearranged Table 1 and split it into three. The three tables each deal with one general topic. And we have also provided a new carton figure to illustrate the signaling pathways in different major immune cells. In addition, we have revised the presentation of signalling to make it more amiable.

Comment 1: Line 40: add "which" after "vesicles"              

Response: “which” is added (line 41).

Comment 2: Lines 59-60: Granule mediated damage within phagosomes / phagolysosomes is a major mechanism of control of intracellular pathogens, including fungi. Therefore, I am curious to understand why, given the broad title of the review, why the authors chose to exclude this mode from their discussion.

Response: During the initial planning of the manuscript, we had considered this possibility, but chose to focus on degranulation, which is the extracellular release of secretory granules. There are many intracellular functions of granules, which include phagosomal-lysosomal fusion. We felt it was better to focus on released granules, and not make the review potentially unwieldly by adding a discussion of intracellular functions.

Comment 3: Line 69: change to the plural form "scientists" here.

Response: we have changed it from its singular to plural “scientists” (line 70).

Comment 4: The formatting of Table 1 makes it very difficult to read. Is it possible to align each cell's text at the vertical top (instead of the current vertical middle)? Also, it might make it easier to read with a slightly smaller font size and/or placement in the landscape format. Additionally, since the table spans multiple pages, is it possible to repeat the header row at the top of each page, so as to keep the columns in context?

Response: We thank the reviewer for this comment, which has allowed is to improve the presentation. We have split Table 1 into 3 separate tables to align it with topics presented in text. We have made other editing to make the tables easy to follow. Table 1. Granule types and contents in various immune cells (line 114). Table 2. Pathways and modes of degranulation in various immune cells (line 220). Table 3. The roles of microtubules, dyneins, kinesins, and SNAPs/SNAREs in granule trafficking in various immune cells (line 255).

Comment 5: Granule Characteristics: I think it is necessary to ensure that the information already mentioned in Table 1 is not repeated in text form in this section.

Responses: We understand that there are some redundancies in the tables and text, however we felt that the tables and are there to support the text and chose to present details in the text so that they are not so simple to make them less useful. In order to reduce redundancy, we have removed the references from the table, which should also simplify the presentation.

Comment 6: Section 5 on signal transduction: By this point, I am wondering if there isn't a better way to describe the signal-to-degranulation pathways without using a sea of text. Is it possible to do a diagram(s), or schema(s), showing the network of pathways?

Response: We agree with the reviewer’s comments. We have inserted a cartoon, which we hope will help readers grasp the similarities and differences in the pathways of various immune cells (line 224).

Comment 7: Section on Mast cells: I think it is important to clarify the differences between the two processes by which mast cells respond with degranulation to allergens and to pathogens.

Response: We have further clarified the difference between mast cell activation during allergic response and fungal infection by adding the sentence “activation of mast cells during allergic responses occurs through IgE receptor-ligand interaction; however, activation is achieved through other receptors such as TLR4 and CCR1 against fungi” (lines 438-440)

Reviewer 2 Report

Many articles address the issue of the immune response to fungal infections. The originality of the subject's approach in this review was to describe in detail the degranulation of immune cells, which is extremely interesting and useful for understanding the immune response. An interesting and well written work. The article is well structured, complete and complex, yet easy to follow. I would have a small suggestion for the authors: since during the article reference is made to various types of fungi, I think that in the introduction, in the classification (line 39), where the large classes of fungi are mentioned, some examples of fungi could be given. which are or may become pathogenic to humans for each of the three major classes.

Author Response

General comment: The originality of the subject's approach in this review was to describe in detail the degranulation of immune cells, which is extremely interesting and useful for understanding the immune response. An interesting and well written work. The article is well structured, complete and complex, yet easy to follow.

Response: We thank the reviewer for this encouraging comment.

Comment 1: “…, in the classification (line 39), where the large classes of fungi are mentioned, some examples of fungi could be given. which are or may become pathogenic to humans for each of the three major classes.”

Response: At the reviewer’s request, we have included examples of pathogens in the major phyla (lines 39-40).

Round 2

Reviewer 1 Report

Thank you including the new figure. That and the reformatting of the tables have improved the presentation of this review greatly. I'm recommending acceptance.